# `Python-JAX`-based Fast Stokesian Dynamics

**Kim William Torre[1]⋆, Raoul D. Schram[2]‡, and Joost de Graaf[1]†**

**1** Institute for Theoretical Physics, Center for Extreme Matter and Emergent Phenomena, Utrecht University, Princetonplein 5, 3584 CC Utrecht, The Netherlands
**2** Research and Data Management Services, Information and Technology Services, Utrecht University, Heidelberglaan 8, 3584 CS Utrecht, The Netherlands

⋆ k.w.torre@uu.nl , ‡ r.d.schram@uu.nl , † j.degraaf@uu.nl

## Abstract

**Stokesian Dynamics (SD) is a powerful computational framework for simulating the motion of particles in a viscous Newtonian fluid under Stokes-flow conditions. Traditional SD implementations can be computationally expensive as they rely on the inversion of large mobility matrices to determine hydrodynamic interactions. Recently, however, the simulation of thermalized systems with large numbers of particles has become feasible [Fiore and Swan, J. Fluid. Mech. 878, 544 (2019)]. Their "fast Stokesian dynamics" (FSD) method leverages a saddle-point formulation to ensure overall scaling of the algorithm that is linear in the number of particles $\mathcal{O}(N)$; performance relies on dedicated graphics-processing-unit computing. Here, we present a different route toward implementing FSD, which instead leverages the Just-in-Time (JIT) compilation capabilities of `Google JAX`. We refer to this implementation as JFSD and perform benchmarks on it to verify that it has the right scaling and is sufficiently fast by the standards of modern computational physics. In addition, we provide a series of physical test cases that help ensure accuracy and robustness, as the code undergoes further development. Thus, JFSD is ready to facilitate the study of hydrodynamic effects in particle suspensions across the domains of soft, active, and granular matter.**

# 1 Introduction

The study of hydrodynamic interactions in particle suspensions has attracted considerable interest in colloid science and fluid mechanics over the past decades [2–4]. The development of Stokesian Dynamics (SD) by Brady and Bossis [5] in the late 1980s provided an accurate framework for simulating the many-body hydrodynamic interactions between spherical particles at zero (low) Reynolds number. That is, in the regime where inertia is completely dominated by friction. The Stokes equations that describe the fluid dynamics of such systems are linear. This is leveraged in SD to divide the mobility problem into a far- and near-field component. The split enables the precise modeling of long-range, many-body hydrodynamic interactions and short-range, lubrication forces [6–8], both of which can strongly influence the behavior of colloidal suspensions [9–19].

However, despite its accuracy, SD did not become as widely adopted as other hydrodynamics methods for particle suspensions [20]. Among these, lattice-based methods for computational fluid dynamics gained considerable traction, including lattice-Boltzmann (LB) [21–25], multi-particle collision dynamics (MPCD) [26, 27] — or stochastic rotation dynamics (SRD) [28] — and Fluid Particle Dynamics (FPD) [29].

Contrasting the features of these methods, we see the following advantages for SD. The algorithm relies on multipole expansion and pairwise approximations to compute hydrodynamic interactions without directly solving for the fluid flow. This means that at low dilution, there is an intrinsic advantage to using SD. However, LB and MPCD are particularly effective for determining fluid-structure interactions involving complex geometries [21–25]. Yet, achieving the same level of accuracy in both many-body and lubrication interactions as SD requires the use of prohibitively fine resolution in LB, leading to higher computational demands [24, 30]. MPCD and SRD operate on similar principles [26–28]; however, they are intrinsically stochastic, with control parameters such as viscosity emerging as collective properties of the system. SD instead provides explicit control over thermal and athermal motion, like LB. Lastly, FPD, as introduced by Tanaka and Araki [29], directly solves the Navier-Stokes equation on a lattice. It treats colloidal particles as regions of higher viscosity fluid with smooth interfacial profiles. This eliminates the need to explicitly resolve solid-fluid boundaries, but it introduces approximations. These depend on the viscosity ratio between the particles and surrounding fluid, as well as on the lattice resolution.

The major bottlenecks for the use of SD were: (i) the computational cost involved in propagating the dynamics and (ii) that it is challenging to implement the algorithm correctly. Advancements following the original implementation [5] have therefore focused on improving SD's computational performance [31–33]. These include the incorporation of iterative solvers and preconditioning strategies. Fiore and Swan overcame some of the last remaining hurdles in making SD truly efficient, through their Fast Stokesian Dynamics (FSD) approach [1]. They combined matrix-free techniques with spectral Ewald methods, which allowed them to further reduce computational costs, improve the scaling to $\mathcal{O}(N)$ overall, and maintain accuracy. They also made their algorithm widely available through integration in the HOOMD [34] simulation package. This means that, provided that complex geometries are not critical to the analysis of a system, FSD provides a competitive alternative to more widely used lattice-based approaches.

However, the efficiency obtained in FSD was limited to `NVIDIA`® graphics processing units (GPUs), through an internal reliance on `CUDA`® programming.

In this paper, we present an alternative route toward making FSD more widely available and cross-platform compatible. We have ported the FSD algorithm to a `Python`-based framework that relies on the `Google JAX` library [35]. We name this version of SD, `Python–JAX` Fast Stokesian Dynamics, or `Python-JFSD` for short. Our implementation ensures compatibility with contemporary `CUDA`® and `CuDNN`® libraries. It also addresses limitations in the original code caused by updates to the `HOOMD` application programming interface (API). `Python`-JFSD further offers a user-friendly interface and provides flexibility in choosing between periodic and open boundary conditions for hydrodynamic interactions. By leveraging `Google JAX`'s Just-In-Time (JIT) compilation [35], the framework supports the efficient addition of new features, ensuring modularity and extensibility.

The remainder of this paper is organized as follows. In Section 2, we first provide a brief overview of the FSD method. Section 3 contains the results of a set of physical unit test that probe the implementation accuracy and benchmark its performances. In Section 4, we discuss the applicability range of JFSD, as well as its strong and weak points. We close with a summary and outlook in Section 5.

## 2 Method

Consider a system comprising $N$ spherical particles of diameter $\sigma$, suspended in a Newtonian solvent with viscosity $\eta$. We have a prescribed background flow $\mathcal{U}^\infty$, which is a $6N$-dimensional vector containing the linear and angular fluid velocities at the position of each particle. We assume viscous forces dominate inertial ones, such that the Reynolds number $\mathrm{Re} \ll 1$, and the particles undergo over-damped motion. Under these conditions, the particle dynamics are prescribed by a linear system of equations that couple the dynamical degrees of freedom (forces and torques $\mathcal{F}$, and stresses $\mathcal{S}$) to kinematic degrees of freedom (linear and angular velocities $\mathcal{U}$ and strain rates $\mathcal{E}$) via the grand-resistance matrix $\mathcal{R}$. The relation is given by

$$\begin{pmatrix} \mathcal{F}^H \\ \mathcal{S}^H \end{pmatrix} = -\mathcal{R} \cdot \begin{pmatrix} \mathcal{U} - \mathcal{U}^\infty \\ -\mathcal{E}^\infty \end{pmatrix}, \tag{1}$$

where $\mathcal{U}$ is a $6N$-dimensional vector containing the linear and angular velocities of all particles, $\mathcal{E}^\infty$ is the strain rate of the background flow, $\mathcal{F}^H$ denotes the hydrodynamic drag forces and torques, and $\mathcal{S}^H$ is a $5N$-dimensional vector of the independent components of the hydrodynamic stresslets.

The generalized velocities of all $N$ particles can then be expressed as

$$\mathcal{U} - \mathcal{U}^\infty = \mathbf{R}_{\mathcal{F}\mathcal{U}}^{-1} \cdot (\mathcal{F}^\mathcal{P} + \mathbf{R}_{\mathcal{F}\mathcal{E}} \cdot \mathcal{E}^\infty) + \sqrt{\frac{2k_\mathrm{B}T}{\Delta t}} \mathbf{R}_{\mathcal{F}\mathcal{U}}^{-1/2} \cdot \psi + k_\mathrm{B}T \; \nabla \cdot \mathbf{R}_{\mathcal{F}\mathcal{U}}^{-1}. \tag{2}$$

Here, $\mathcal{F}^\mathcal{P}$ is a $6N$-dimensional vector whose first $3N$ components are the forces applied to the particles, and the last $3N$ components are the applied torques. The vector $\psi$ is a $6N$-dimensional vector containing random variables normally distributed with zero mean and unit variance, and $\Delta t$ is the simulation timestep. The term $k_\mathrm{B}T$ represents the thermal energy; $k_\mathrm{B}$ is the Boltzmann constant and $T$ the temperature. The tensor $\mathbf{R}_{\mathcal{F}\mathcal{U}}$ is the upper-left subblock of $\mathcal{R}$, which maps velocities to forces, while $\mathbf{R}_{\mathcal{F}\mathcal{E}}$ is the upper-right subblock that couples strain rates to forces in $\mathcal{R}$, respectively. In Eq. (2), the first two terms represent deterministic contributions to the particle dynamics coming from the applied forces and torques, as well as the background flow, while the last two terms account for thermal fluctuations.

In SD [5], hydrodynamic interactions are incorporated through a combination of the mobility and resistance frameworks. The construction of the grand resistance matrix $\mathcal{R}$ begins with the far-field mobility matrix $\mathcal{M}^{\mathrm{ff}}$. Short-range interactions are resolved in a pairwise manner by adding near-field resistance contributions $\mathcal{R}^{\mathrm{nf}}$ [6, 8]. To avoid double-counting, the far-field pairwise contributions $\mathcal{R}^{\mathrm{ff}}_{\mathrm{2B}}$ are subtracted

$$\mathcal{R} = (\mathcal{M}^{\mathrm{ff}})^{-1} + \mathcal{R}^{\mathrm{nf}} - \mathcal{R}^{\mathrm{ff}}_{\mathrm{2B}}. \tag{3}$$

The FSD method [1] extends the SD framework by formulating the particle dynamics as a linear system of $17N$ equations using a saddle-point matrix that integrates near-field (short-ranged, pairwise additive) and far-field (long-ranged, many-body) interactions. This formulation enables iterative solvers with specialized preconditioners to efficiently handle the resulting linear systems. The approach avoids the explicit inversion of ill-conditioned hydrodynamic operators and drastically reducing computational costs. Brownian forces are computed by combining the positively-split Ewald method [36] and an iterative Krylov subspace method [37], both seamlessly integrated into the saddle-point formulation. We refer the interested reader to Ref. [1] for the full details.

Our implementation of the method, JFSD, is structured as a modular Python package optimized for scalability and high-performance hydrodynamic simulations. In particular, the configuration management module leverages a user-friendly TOML file that specifies simulation parameters — such as the number of steps, particle count, time step, initialization settings, and output options — providing a transparent and easily modifiable interface for setting up simulations. The modular design allows for easy extension of pairwise interaction models, enabling users to implement custom hydrodynamic kernels or introduce *ad hoc* force laws with minimal modifications. The framework also provides built-in support for steady and oscillatory shear flows, with a straightforward pathway toward incorporating more complex shear protocols by modifying existing velocity gradients. Boundary conditions can be adjusted efficiently by altering the scalar mobility functions, as demonstrated in the already implemented transition between open and periodic boundary conditions. Lastly, our neighbor list management was derived from the one used in JAX-MD [38], a widely used molecular dynamics framework. Our implementation ensures FSD-compatible handling of particle interactions in both dense and dilute systems. Full details of our implementation can be found in the documentation accompanying the JFSD release [39].

A key advantage of JFSD is its integration with `Google JAX` [35], which enables Just-In-Time (JIT) compilation and GPU acceleration for significant performance improvements. JAX compiles entire computation graphs into highly optimized routines that eliminate Python overhead by fusing multiple operations into a single efficient kernel. Moreover, its vectorized operations and automatic backend dispatch ensure that computations run optimally on both GPU and CPU, making the framework highly scalable across diverse hardware configurations.

# 3 Method Validations

Following a general discussion of (F)SD, we now show results obtained using JFSD. These are contrasted against two-body analytical expressions and/or independently obtained numerical data. We also showcase the efficiency of the JFSD software by benchmarking the code for the case of a hard-sphere system across a range of volume fractions $\phi$.

## 3.1 Sedimenting Particles

The deterministic mapping from forces to velocities was validated by simulating the sedimentation of three particles aligned along the $x$-axis. Our results are compared to the original

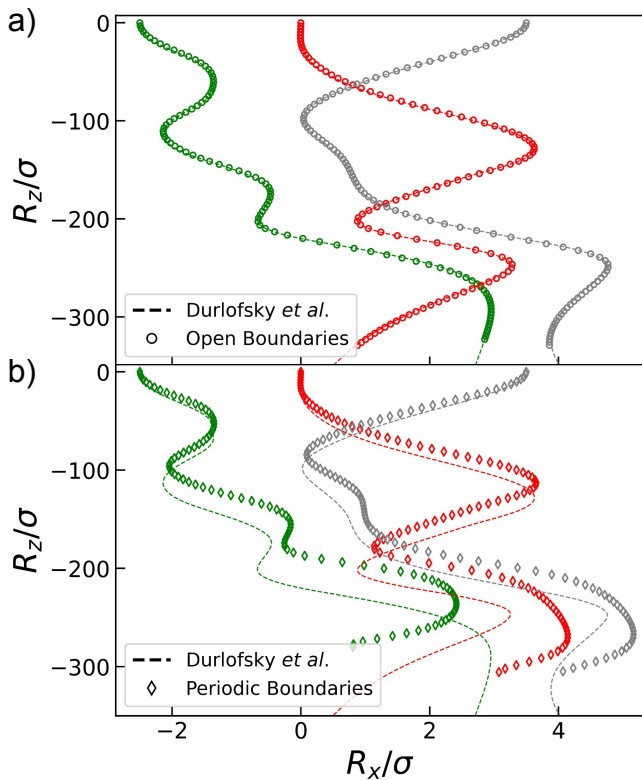

Figure 1: Trajectories of three equal spheres sedimenting vertically, starting from initial positions at $r_1 = (-2.5\sigma, 0, 0)$ (green), $r_2 = (0, 0, 0)$ (red), and $r_3 = (3.5\sigma, 0, 0)$ (gray). Results obtained using the JFSD method under (a) open boundary conditions (circles) and (b) periodic boundary conditions (rhombi), in a box of size $L = 25\sigma$. Both panels include reference data from Ref. [40] (dashed curves), where the original data points have been smoothed using spline interpolation.

Stokesian Dynamics (SD) trajectories by Durlofsky *et al.* [40] for an unbounded fluid solvent in Fig. 1. This shows the sedimentation trajectories of the three particles. Here, JFSD demonstrates excellent agreement with the SD reference result. The trajectories aligning closely over the full time interval considered. With JFSD we also considered periodic boundaries, for which discrepancies become pronounced as time progresses. These deviations are expected and arise due to interaction with periodic images. Despite these differences, JFSD faithfully reproduces the initial dynamics. The combined result is sufficient to conclude that JFSD is consistent with the original SD framework.

Next, we computed the sedimentation velocity $U_s$ of a periodic cubic array of $N = 2,197$ hard spheres across a range of $\phi$, see Fig. 2. These results were compared against the benchmark data from Sierou and Brady [31], which revealed that JFSD accurately captures the non-linear decay of $U_s$ with increasing $\phi$. Our implementation gives consistent results even in dense regimes, which implies that it can reliably propagate suspension dynamics across a wide range of concentrations.

## 3.2 Hydrodynamic Effects under Shear Flow

Now that we have examined the quality of the force-torque and (angular) velocity coupling, we turn to shear. We investigated the behavior of particle pairs in simple shear flow to bench-

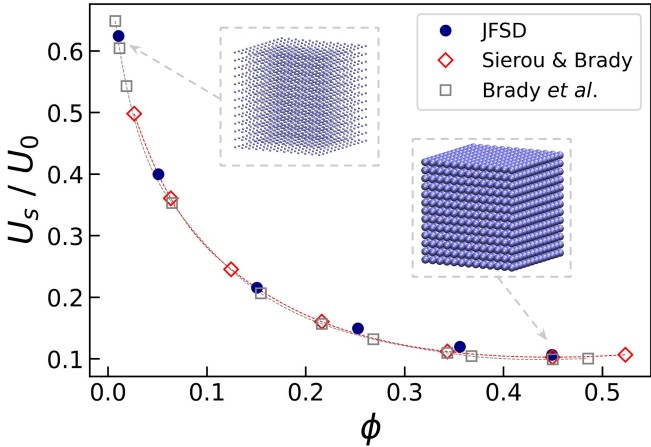

Figure 2: Comparison of sedimenting velocity $U_s$ (blue dots) for a simple cubic array of $N = 2{,}197$ particles (insets show examples) as a function of the volume fraction $\phi$. The sedimentation velocity is normalized by the infinite dilution limit $U_0$. Results from the JFSD method under periodic boundary conditions are contrasted with those obtained using Sierou's accelerated Stokesian dynamics [31] (red rhombi) and Brady *et al.*'s original Stokesian dynamics [41] (gray squares). The curves connecting the two reference data sets, obtained through spline interpolation, serve to guide the eye.

mark the accuracy of JFSD under these conditions, see Fig. 3, which shows three example trajectories. The results align closely with analytical predictions [4, 42], demonstrating JFSD's performance.

In addition to pair interactions, we computed the shear viscosity $\eta_{\text{eff}}$ of a periodic simple cubic array of (again) $N = 2{,}197$ hard spheres as a function of $\phi$. Figure 4 compares our result with data from the original FSD implementation by Fiore and Swan [1]. The accurate agreement shows that both algorithms produce the same results and are capable of computing rheological properties even at high $\phi$.

Our combined shear results allow us to conclude that this part of the algorithm is correctly implemented. Of course, additional testing was performed on the behavior of the $\mathcal{R}$ matrix, but we chose only to report on physical observables here.

## 3.3 Thermal Motion

Having verified the deterministic part of the algorithm, we turned our attention to thermal fluctuations. The mean squared displacement (MSD) $\langle r^2 \rangle$ of a single particle in 3D was computed under periodic boundary conditions and benchmarked against Hasimoto's corrected expression for periodic boundaries [43]. To generate the MSD curves in Fig. 5, we simulated for 100 Brownian times and repeated the process for 10 independent samples for each box size $L$, averaging the results across these runs. The figure reveals that the simulated MSD aligns closely with theoretical predictions, confirming the accurate implementation of periodic images and the validity of thermal diffusion in the JFSD framework.

However, MSD analysis alone is not sufficient to validate the thermal part of the algorithm. This is because in SD, there is a 'Brownian drift' term that is required to ensure stationarity under the Gibbs–Boltzmann distribution and generate particle configurations with the correct statistics at equilibrium [1, 44]. To demonstrate that we have correctly implemented this aspect, we draw inspiration from a test originally considered by Fiore and Swan [1, 44]. That is, we simulated a pair of particles, in an unbounded three-dimensional (3D) fluid, interacting

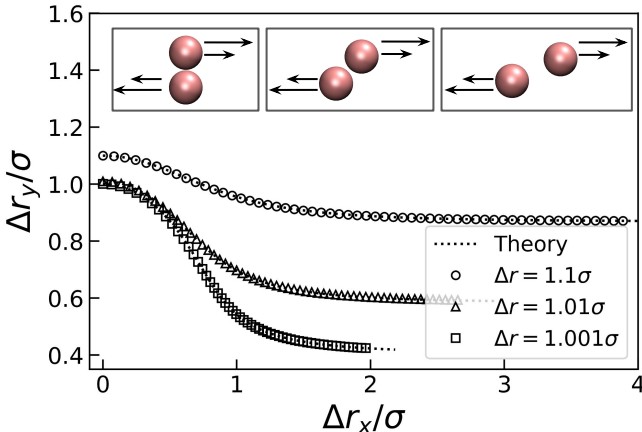

Figure 3: Trajectories of a particle pair in simple shear flow under open boundary conditions. Results from JFSD (symbols) are compared against analytical predictions [4, 42] (dotted lines) for three initial center-to-center separations $\Delta r$. These trajectories are represented in the distance along the $x$- and $y$-axes, respectively, normalized by the particle diameter $\sigma$. The insets show three snapshots of a trajectory for spheres spaced $\Delta r = 1.01\sigma$ apart orthogonal to the shear plane.

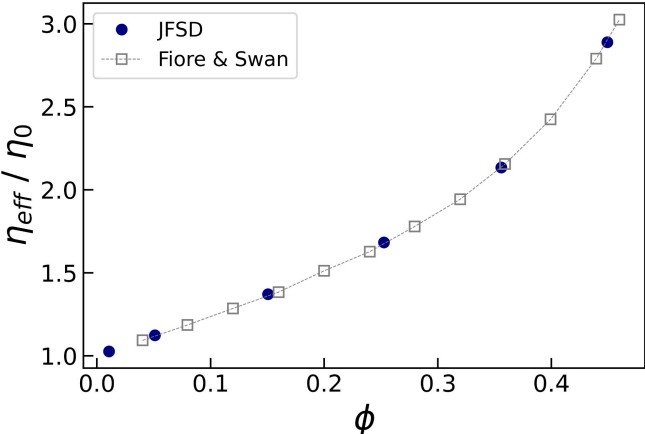

Figure 4: Shear viscosity $\eta_{\text{eff}}$ of a simple cubic array of $N = 2,197$ particles as a function of volume fraction $\phi$. The results are normalized by the bare viscosity of the suspending medium $\eta_0$. Python-JFSD results (blue circles) are compared with data from Fiore and Swan [1] (open squares). The dotted connecting curve serves to guide the eye.

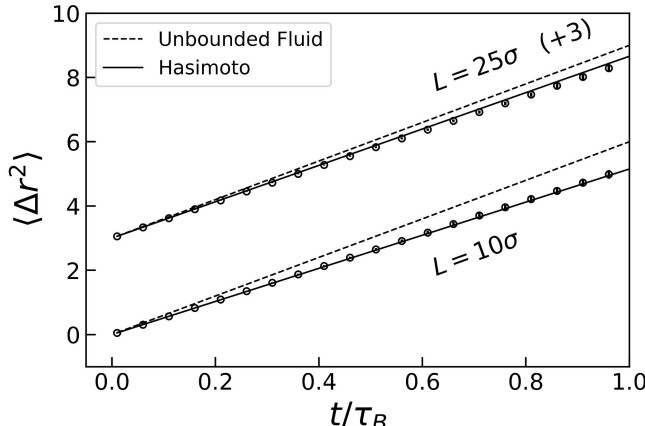

Figure 5: Mean squared displacement $\langle r^2 \rangle$ of a single particle as a function of time $t$ divided by the Brownian time $t_{\mathrm{B}}$. The JFSD data is given by open circles with error bars representing the standard error of the mean for two box sizes $L$ as labelled. The data for the $L = 25\sigma$ ($\sigma$ is the sphere diameter) box is shifted by the constant offset $+3$ in the vertical direction to help aid the presentation. The dotted line shows the analytic $\langle r^2 \rangle = 6Dt$ result for an unbounded fluid, while the solid line applies Hasimoto's correction to obtain the appropriate expression for a periodic domain [43].

*via* a linear potential

$$V(r) = k|r - r_0|. \tag{4}$$

Here, $k$ is the potential strength, $r$ is the radial distance between the sphere centers, and $r_0$ is the equilibrium distance. Figure 6 compares the average distribution of radial distances $p(r)$ — derived from 5 independent simulations — with duration of 5 Brownian times each — against the theoretical Boltzmann probability density function [1]. The good agreement demonstrates Python-JFSD's ability to reproduce expected equilibrium distributions. Small deviations in the tail of the distribution arise primarily from the accumulation of numerical errors in long trajectories, due to the use of a first-order Euler integration scheme [45].

We also considered large particle number systems ($N = 2,500$) with periodic boundary conditions. For these, we computed the short-time self-diffusion coefficient $D_s$ as a function of $\phi$ for hard-sphere suspensions, see Fig. 7. Our results are compared therein with two independently derived datasets. The first is by Ladd [46] and was obtained from a multipole-moment expansion, combined with pairwise lubrication interactions, that contains a higher number of moments than conventional SD. The second follows from Sierou's and Brady's work [31] and represents the Accellerated Stokesian Dynamics method. Both datasets were obtained for open systems, hence we have corrected for the periodicity effects using Hasimoto's scaling factor [31, 43, 46]. The close agreement between our results and the (scaled) literature values confirms JFSD's ability to resolve thermal fluctuations in systems with many particles in periodic boundary conditions.

## 3.4 Algorithmic Performance and Scaling

We assessed the computational performance of Python-JFSD using an NVIDIA GeForce RTX 4060 Ti with 8GB of VRAM. As a standard performance metric, we define the particle time steps per second (PTPS) as PTPS $= N/T_{\mathrm{step}}$, where $T_{\mathrm{step}}$ is the clock time in seconds required to perform a single simulation step as a function of the number of particles $N$. The setup consists of $N$ hard spheres arranged in a cubic array, interacting only via excluded-volume

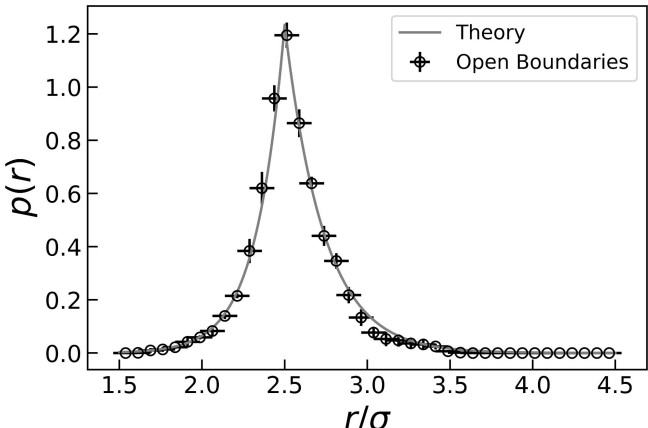

Figure 6: Radial distribution $p(r) \propto r^2 \exp(-V(r)/k_\mathrm{B}T)$ of a pair of particles inter-acting *via* a linear potential (see Eq. (4)) under open boundary conditions, with $r$ the the center-to-center distance between the two spheres. The JFSD data is shown using open circles, with the vertical error bars showing the standard error of the mean (computed from 5 independent samples). The horizontal error bars follow from the fact that the data was obtained by a binning procedure.

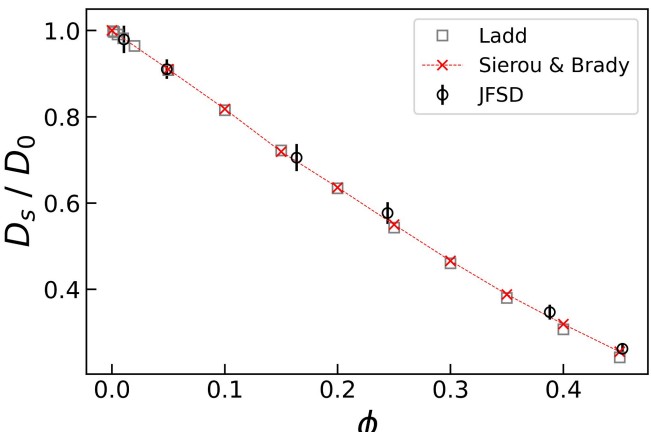

Figure 7: Short-time self-diffusion coefficient $D_s$ for hard-sphere suspensions under periodic boundary conditions as a function of volume fraction $\phi$. The diffusion coefficient is normalized by the bulk value $D_0$. Results from `Python-JFSD` (open circles) have error bars that show the standard error of the mean. These are compared to the work by Ladd [46] (open circle) and Sierou [31] (red crosses). Both datasets are corrected for finite-size effects *via* the Hasimoto factor [43] and the dashed curves help guide the eye.

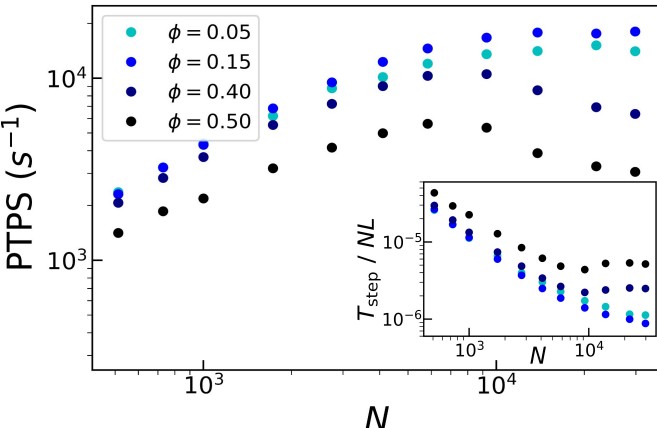

Figure 8: Particle time steps per second (PTPS) as a function of the number of particles $N$ in a periodic simulation volume. Four volume fractions were considered: $\phi \in \{0.05, 0.15, 0.4, 0.5\}$, as labeled. At low $\phi$, PTPS plateaus for large $N$, while for high $\phi$, PTPS decreases with $N$ due to system-size dependence. The inset shows the normalized step time per particle per unit box size, $T_{\text{step}}/(NL)$, which remains constant for large $\phi$, confirming that the algorithm maintains linear scaling despite the decrease in PTPS in the main plot.

interactions to prevent overlap. The particles undergo thermal motion, and a simple shear flow is applied in the $xy$-plane (with vorticity in the $z$-direction, flow in the $x$-direction, and flow gradient in the $y$-direction). Periodic boundary conditions are imposed. To ensure robust benchmarking, the system is simulated for 50 steps, and the average $T_{\text{step}}$ is computed after excluding the first step, which involves JIT compilation and can take up to 1–2 minutes.

Since computational performance depends on the volume fraction $\phi$, we considered four representative values, $\phi \in \{0.05, 0.15, 0.4, 0.5\}$, to assess scaling behavior. This dependence arises from the iterative solver used in the FSD algorithm [1], where the number of iterations required for convergence increases due to lubrication interactions, which become more significant at higher $\phi$ as particles are closer together. The results, shown in Fig. 8, present PTPS as a function of $N$.

The PTPS trends indicate that at low $\phi$, performance reaches a steady plateau, as expected for sufficiently large $N$. Conversely, at higher volume fractions, PTPS continues to decrease with increasing $N$. This behavior suggests that system size influences solver convergence, which introduces additional computational overhead. The inset in Fig. 8 clarifies this effect: for large $\phi$, the normalized time step per particle per unit box size remains unchanged, verifying that the scaling of the FSD algorithm is preserved. This shows that although PTPS decreases at high $\phi$, the core computational efficiency of the method remains intact, consistent with theoretical predictions [1].

## 4   Discussion

SD is a well-established framework for simulating particle suspensions [1, 5, 31–33, 40], capable of accurately resolving both many-body hydrodynamic interactions and near-field lubrication forces. It is particularly effective for systems requiring precise particle-level hydrodynamic modeling without complicated boundary conditions. SD relies on direct calculations of forces and torques, which allows for computational efficiency without the need for fine spatial dis-

cretization. Recent applications include sedimentation in colloidal suspensions [13], colloidal gelation [18], and the dynamics of active matter [47] in confining geometries [48,49]. These examples showcase SD's versatility across a broad range of (colloidal) particulate systems.

Implementing SD methods, however, presents significant challenges due to the algorithm's reliance on complex mathematical algorithms [50] and the precision required to accurately model lubrication forces and many-body interactions. Ensuring correctness and robustness needs meticulous validation. To address this, our JFSD implementation incorporates continuous integration workflows, automatically checking each pull request – originating from Git-based version control – against an extensive suite of benchmarks [39]. This rigorous testing, combined with a modular design, allows users to choose between periodic or open boundary conditions and various hydrodynamic models, including FSD, which has been the focus of this paper. In addition to these features, our software suite is also capable of modeling hydrodynamic interactions at the Rotne-Prager-Yamakawa level [36,51,52], which is suitable for capturing pair-wise, far-field coupling. We have further implemented a simple Brownian Dynamics code, which focuses on the hydrodynamic self-interaction only. However, this last algorithm has not been optimized to the same degree as is available in dedicated software packages, as our primary focus has been on realizing a new version of FSD.

JFSD was created to provide greater accessibility to SD by leveraging `Python`'s simplicity and interoperability, while maintaining the competitive computational performance of dedicated implementations. Thanks to `Google JAX`'s JIT compilation and reliance on GPU acceleration, our software achieves speeds comparable to previous $CUDA^{\circledR}$ implementations [1]. Presently, JFSD is capable of simulating approximately 30,000–50,000 particles within the memory constraints of mid-range GPUs. While this imposes a practical upper limit, ongoing developments in `Google JAX`, particularly the support for multi-core CPU and distributed computing, could further expand the range of the software in the future.

## 5   Conclusion and Outlook

In summary, we have presented a `Python`-based implementation of fast Stokesian Dynamics that relies on the `Google JAX`, which we refer to as JFSD. Our work addresses the use issues posed by the currently outdated $CUDA^{\circledR}$ code base [1] originally developed by Fiore and Swan. By utilizing Just-In-Time compilation [35], JFSD achieves identical scaling, while maintaining performance and offering a user-friendly and modular interface. Our software is capable of simulations involving tens of thousands of particles on a modern desktop GPU.

We built upon the extensive literature for SD and hydrodynamic interactions in suspensions to validate the code. Key benchmarks performed in this paper include: sedimentation, shear rheology, and thermal motion. We found strong agreement with the literature, demonstrating the quality of our implementation. The inclusion of both open and periodic boundary conditions, along with alternative hydrodynamic models, ensures flexibility and broad applicability for diverse physical systems. JFSD is thus ready to enable new research into the effect of hydrodynamic interactions in particle suspensions.

## Acknowledgements

We developed JFSD with the aim of giving back to the fluid-dynamics community, whose efforts and developments [1,34,36,40,44,52,53] helped support our own research [18,54]. We hope that our implementation of FSD will allow researchers to continue to make use of the excellent algorithms developed by Fiore and Swan [1] for many years to come. Thank you Jim for the

kindness that you showed us. We are also grateful to Andrew M. Fiore, Zhouyang Ge, and Gwynn Elfring; Luca Leone and Henri Menke; and Athanasios Machas and Dimos Aslanis for useful discussions, which helped us significantly improve our understanding of FSD, `Google JAX`, and the user/developer perspective, respectively.

**Data availability statement**  An open data package containing the means to reproduce the results of the simulations is available at: DOI. The specific version of JFSD used to generate the results in this paper is tagged as `v0.2.0`, which will aid with reproducibility.

**Author contributions**  Conceptualization, J.d.G. & K.W.T.; Methodology, K.W.T. & R.S.; Numerical calculations, K.W.T., Validation, K.W.T. & R.S.; Investigation, K.W.T.; Writing — Original Draft, K.W.T.; Writing — Review & Editing, J.d.G.; Funding Acquisition, J.d.G.; Resources, J.d.G.; Supervision, J.d.G.

**Funding information**  The authors acknowledge the Dutch Research Council NWO for funding through OCENW.KLEIN.354, as well as the International Fine Particle Research Institute for funding through collaboration grant CRR-118-01.

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
