# Peer review of "Python-JAX-based Fast Stokesian Dynamics"

_SciPost Physics Codebases, doi:SciPost Phys. Codebases 56-r0.2 (2025) , SciPost Phys. Codebases 56 (2025)_

## Round 1 · Referee Report · Anonymous (Referee 1) · 2025-4-7

Strengths

1- Provides a new, faster implementation of Stokesian Dynamics for low Reynolds number flows of suspensions 2- Thorough validation of correct performance against existing literature 3- Good benchmarking for speed against current state of the art 4- Clever use of JIT compilation which scales well for large systems (though it is poor for small systems because of the slow initial compilation step)

Weaknesses

1- Small grammatical errors - a few sentences that are not sentences (e.g. page 1, line 5; page 5, line 3) 2- Could add plans for extending to planar extensional flow (using Kraynik-Reinelt boundary conditions) which would strengthen the software considerably 3- In section 3, I would welcome more discussion of why box size affects the performance of the code 4- Formatting of references is poor (many missing capital letters in article titles)

Report

This is a lovely paper. It presents very clearly why Stokesian Dynamics is still a method worth using, and provides a new, faster implementation. Everything is properly validated and benchmarked, and example applications provided in detail. As far as I can tell this is being made available in a coherent and easy-to-access way, though that is not my area of expertise.

Requested changes

1- Correct small grammatical errors at page 1, line 5; page 5, line 3 2- In section 3, explain why box size affects the performance of the code 3- Correct formatting of references

Recommendation

Publish (easily meets expectations and criteria for this Journal; among top 50%)

---

## Editorial Decision

published